# Has a change in established care pathways during the first wave of the COVID-19 pandemic led to an excess death rate in the fragility fracture population? A longitudinal cohort study of 1846 patients

Adeel Ikram ![ORCID],[1,2] Alan Norrish ![ORCID],[1,3] Luke Ollivere,[2] Jessica Nightingale ![ORCID],[1,2] Ana Valdes ![ORCID],[4,5] Benjamin J Ollivere ![ORCID][1,2,5]

For numbered affiliations see end of article.

**Correspondence to**
Mr Adeel Ikram;
adeel.ikram@nottingham.ac.uk

## ABSTRACT

**Objective** During the first wave of the COVID-19 pandemic, changes to established care pathways and discharge thresholds for patients with fragility fractures were made. This was to increase hospital bed capacity and minimise the inpatient risk of contracting COVID-19. This study aims to identify the excess death rate in this population during the first wave of the pandemic.

**Design** A longitudinal cohort study of patients with fragility fractures identified by specific International Classification of Diseases (ICD)-10 codes. The first wave of the pandemic was defined as the 3-month period between 1 March and 1 June 2020. The control group presented between 1 March and 1 June 2019.

**Setting** Two acute National Health Service hospitals within the East Midlands region of England.

**Participants** 1846 patients with fragility fractures over the aforementioned two specified matched time points.

**Primary and secondary outcome measures** Four-month mortality of all patients with fragility fractures with a subanalysis of patients with fragility hip fractures.

**Results** 832 patients with fragility fracture were admitted during the pandemic period (104 diagnosed with COVID-19). 1014 patients presented with fragility fractures in the control group. Mortality in patients with fragility fracture without COVID-19 was significantly higher among pandemic period admissions (14.7%) than the pre-pandemic cohort (10.2%) (HR=1.86; 95% CI 1.41 to 2.45; p<0.001) adjusted for age and sex. Length of stay was shorter during the pandemic period (effect size=−4.2 days; 95% CI −5.8 to −3.1, p<0.001). Subanalysis of patients with fragility hip fracture revealed a mortality of 8.4% in the pre-pandemic cohort, and 15.48% during pandemic admissions with no COVID-19 diagnosis (HR=2.08; 95% CI 1.11 to 3.90; p=0.021).

**Conclusions** There is a significant increase in excess death, not explained by confirmed COVID-19 infections. Altered care pathways and aggressive discharge criteria during the pandemic are likely responsible for the increase in excess deaths.

## STRENGTHS AND LIMITATIONS OF THIS STUDY

⇒ This study is one of the largest cohort studies in patients with fragility hip fracture investigating the excess mortality in COVID-19-negative patients during the first wave of the COVID-19 pandemic.
⇒ Specific International Classification of Diseases (ICD)-10 codes allowed us to identify the same injury patterns over both groups and a comparative time matched cohort was used as a control.
⇒ The effect of the provision of personal protective equipment and transmission between healthcare staff and patients was not known and these data were not available.
⇒ The redeployment of support staff particularly physiotherapists and orthogeriatric staff for patients with fragility fracture may have had an influence on negative outcomes; however, specific redeployment data were not available for this study.

## INTRODUCTION

The Office for National Statistics in the UK states between 1 March and 30 June 2020, there were 218 837 deaths in England and Wales, of which 50 335 (23%) involved COVID-19.[1] At the start of March 2020, the number of deaths per day was below the 5-year average, possibly due to mild winter levels of circulating influenza. By the end of the first week of April 2020, this figure was more than double the 5-year average, demonstrating the significant impact of COVID-19 on all-cause mortality.[1]

The first wave of the COVID-19 pandemic transformed the delivery of orthopaedic services. There was cessation of elective operative practice, a reduction in trauma operative capacity as intensive care bed capacity needed to expand into theatres, and redeployment

of all grades of orthopaedic staff (surgeons, medics and allied healthcare professionals) to frontline emergency and intensive care. Established evidence-based care pathways were modified to reduce length of stay where possible and to lower discharge thresholds to increase bed capacity within hospitals. Following the imposition of a national lockdown, the government advice to the public was to work from home and undertake essential travel only. Following this, there was a reduction in the activity of the general population and therefore a reduction in the presentation of major and polytrauma activity. However, the number of patients presenting with fragility hip fractures did not cease. The WHO stated in April 2020 that elderly patients are at highest risk of COVID-19,[2] and a recent study demonstrated an increased risk of mortality of 7.8% of people aged over 80 years with an age-related gradient.[3]

Studies have shown an increase in excess deaths by analysis of all-cause mortality, not just due to COVID-19 infection, during the first wave of the global COVID-19 pandemic.[4] Non-COVID-19 excess deaths are associated with increasing age, with the largest increases in non-COVID-19 deaths being attributed to dementia and frailty. It is well known patients who present with fragility fractures are at risk of poorer outcomes,[5 6 1 2] and outcomes are care pathway dependent.

This study was designed to quantify the excess mortality related to the pandemic in this group following the modification of established care pathways in the light of the first wave. By maintaining long-established evidence-based care pathways for fragility fractures, even in the face of a future wave of the COVID-19 pandemic or future pandemics, excess death in this vulnerable population may be reduced.

## MATERIALS AND METHODS

A longitudinal cohort study design identified study and control groups from the same institution, a UK hospital group with a major trauma centre and 1700 inpatient beds. This was an analysis of prospectively collected data for both the control group and study group with data in both groups collected during patient admission as part of routine clinical care. The analysis of the control group was performed retrospectively since this was a pre-pandemic period which was used as a comparator. Inclusion criteria for both groups were all patients, of all ages, admitted with an International Classification of Diseases (ICD)-10 code[7] of S72 (fracture of femur), M80 (osteoporosis with current pathological fracture), M96 (intraoperative and post-procedural complications and disorders of post-procedural system, not elsewhere classified), W06 (fall from bed), W19 (unspecified fall) and Y79 (orthopaedic devices associated with adverse incidents). These ICD-10 codes were selected to capture patients who were likely presenting with fragility fractures, and were identified from data entered for the national Hospital Episode Statistics (HES) submission. Patients who present with

fragility fractures have significant medical comorbidities, are elderly and frail. This group of patients are the ones at greatest risk of both COVID-19 and any alterations to established evidence-based care pathways. Two time periods were selected to define groups: (1) the study group were admitted during the first wave of the UK COVID-19 pandemic, between 1 March 2020 and 1 June 2020, and discharged by 30 June 2020; (2) the control group were admitted between 1 March 2019 and 1 June 2019, and discharged by 30 June 2019. The pandemic was declared by the WHO on 11 March 2019, but in the days before this, there was an increase in the number of reported infections in the area served by our institution, so the study period was chosen from 1 March 2019. Excluded were all patients who did not have one of the ICD-10 codes above. Mortality was established using National Health Service Digital data.

Demographic and injury data collected included patient age and sex, the diagnosis, COVID-19 status (as identified in the medical records as 'coronavirus SARS-CoV-2'-positive patients or with a positive reverse transcriptase PCR), number of comorbidities and whether or not the patient had a surgical procedure (as identified through the procedure codes of the submitted HES data). COVID-19 testing was performed on admission for all patients and repeated every 3 days during inpatient stay; this was to facilitate and maintain safe distancing and isolation for COVID-19 contacts. Post-discharge testing was not performed as these patients were either discharged home or to an institution. Patients with hip fracture are not routinely followed up, and those patients with other fragility fractures were advised to not attend follow-up if they developed symptoms or had a positive test.

For the subgroup of patients with hip fracture, prospectively collected data were obtained from the institutional local hip fracture database. On this database, data for every adult patient admitted with a hip fracture are recorded using a modified version of the Standardised Audit of Hip Fractures in Europe data collection form. Data analysed included mechanism of injury, hip fracture type, body mass index (BMI), Abbreviated Mental Test Score (AMTS[8]), mobility status (Fracture Mobility Score[9]), Nottingham Hip Fracture Score (NHFS[10]), haemoglobin concentration (as measured from the full blood count test) on admission and clinical frailty (as recorded on the Rockwood Clinical Frailty Scale[11]). Data collected were confidential and managed in line with national data protection guidelines. Hospital outcome data were available for all patients and included type of surgery, mortality, requirement for invasive or non-invasive ventilation, length of stay and development of complications.

The data were extracted by a data analyst from our local database in July 2020 and ordered in Microsoft Excel (Redmond, Washington, USA) where numerical data comparison between groups was carried out using a Student's t-test assuming a two-tailed distribution with

homoscedastic variance. For categorical data, where a 2×2 contingency table could be used, probability was calculated with the Fisher's exact test. For contingency tables greater than 2×2, the $X^2$ test was used. Statistical significance was set at $p<0.05$. Mortality was calculated from any death of any member of the cohort between 1 March and 29 June in the respective year. The Cox proportional-hazards model was used with STATA V.16.0 (StataCorp, Texas, USA) statistical software package, to compare survival between groups and to calculate the HR where the COVID-19 first wave was the covariate. In addition, Student's t-test (assuming a two-tailed distribution with

homoscedastic variance), Fisher's exact test and $X^2$ test were used to calculate probability.

## Patient and public involvement

Patients or the public were not involved in the design, or conduct, or reporting, or dissemination plans of our research.

## RESULTS

Demographic comparison between groups is shown in table 1A. The percentage of female patients and the

**Table 1** (A) Descriptive characteristics of all patients with fragility fracture presenting during the two study periods; (B) hip fracture subgroup analysis for the two study periods

**A**

| Factor | COVID-19 period group (admitted between 01 Mar 2020 and 01 June 2020) (n=832) | Control period group (admitted between 01 Mar 2019 and 01 June 2019) (n=1014) | P value |
|---|---|---|---|
| Mean age, years (±SD) | 74.8 (±20.0) | 72.8 (±21.4) | 0.045* |
| Female, n (%) | 492/832 (59.1) | 629/1014 (62.0) | 0.213† |
| rtPCR COVID-19 positive, n (%) | 104/832 (12.5) | 0/1014 (0) | <0.001† |
| Undergoing operative procedure, n (%) | 70/832 (8.4) | 765/1014 (75.4) | <0.001† |
| Hip fracture, n (%) | 180/832 (21.6) | 190/1014 (18.7) | 0.129† |

**B**

| Factor | COVID-19 period hip fracture subgroup (total: n=180) | Control period hip fracture subgroup (total: n=190) | P value |
|---|---|---|---|
| Mean age, years (±SD) | 83.2 (±8.8) | 81.9 (±9.2) | 0.183* |
| Female, n (%) | 132/180 (73.3) | 139/190 (73.2) | 1.000† |
| rtPCR COVID-19 positive, n (%) | 12/180 (6.7) | 0/190 (0) | <0.001† |
| Mean BMI, kg/m² (±SD); n | 24.2 (±5.2); 159 | 23.2 (±4.4); 169 | 0.070* |
| Mean AMTS (±SD); n | 6.9 (±3.5); 179 | 6.7 (±3.9); 189 | 0.538* |
| Mobility status, n (available; %) | n=175 | n=187 | |
| ► Freely mobile, no aids | 61/175 (34.9) | 75/187 (40.1) | |
| ► Outside with aids | 107/175 (61.1) | 96/187 (51.3) | 0.076‡ |
| ► Unable to mobilise outside | 7/175 (4.0) | 16/187 (8.6) | |
| Mean Nottingham Hip Fracture Score (±SD); n | 0.082 (±0.063); 172 | 0.081 (±0.054); 181 | 0.838* |
| Anaemia on admission, Hb <12 g/L, n (%) | 36/175 (20.6) | 45/182 (24.7) | 0.378† |
| Hip fracture type, n (%) | | | |
| ► Undisplaced intracapsular | 14/180 (7.8) | 16/190 (8.4) | 0.851† |
| ► Displaced intracapsular | 76/180 (42.2) | 83/190 (43.7) | 0.398† |
| ► Intertrochanteric | 77/180 (42.8) | 79/190 (41.6) | 0.834† |
| ► Reverse oblique or subtrochanteric | 13/180 (7.2) | 12/190 (6.3) | 0.837† |
| RCFS, mean category from 1 to 9 (±SD); n | 4.8 (±1.6); 155 | 4.6 (±1.8); 175 | 0.421* |

*Student's t-test assuming a two-tailed distribution with homoscedastic variance.
†Fisher's exact test.
‡$X^2$ statistic calculated on a 2×3 contingency table.
AMTS, Abbreviated Mental Test Score; BMI, body mass index; Hb, whole blood haemoglobin; RCFS, Rockwood Clinical Frailty Scale; rtPCR, reverse transcriptase PCR.

**Table 2** Outcomes for each group: (A) outcomes for the full cohort; (B) outcomes for the hip fracture subgroup

| A. Outcomes | COVID-19 period group (n=832) | Control group (n=1014) | HR and p value |
|---|---|---|---|
| Mortality, n (%) | 150/832 (18.0) | 104/1014 (10.2) | HR=1.86; 95% CI 1.41 to 2.45; p<0.001 |
| Length of stay, mean days (±SD); n | 8.8 (±8.9); 832 | 12.1 (14.3); 1014 | p<0.001* |

| B. Outcomes | COVID-19 period hip fracture subgroup (n=180) | Control hip fracture subgroup (n=190) | HR and p value |
|---|---|---|---|
| Mortality, n (%) | 30/180 (15.5) | 16/190 (8.4) | HR=2.08; 95% CI 1.11 to 3.90; p=0.021 |
| Mortality adjusted for clinical frailty, n (%) | 30/180 (15.5) | 16/190 (8.4) | HR=2.15; 95% CI 1.15 to 4.04; p=0.016 |
| Length of stay, mean days (±SD); n | 11.1 (±5.5); 180 | 15.3 (±7.4); 190 | p<0.001* |

*Kruskal-Wallis test.
n, number.

number of hip fractures were equivalent between groups. Differences were noted with age and a lower percentage of patients receiving operative treatment during the COVID-19 pandemic.

Hip fractures were the most common diagnoses making up approximately 20% in each of the two comparative groups. These were selected for a more granular subgroup analysis. Table 1B shows the variables used to assess selection bias between the hip fracture subgroups. No differences were seen between subgroups for the variables selected other than prevalence of COVID-19 infection.

Outcomes are recorded in table 2, showing that mortality was significantly higher in the cohort presenting with a fragility fracture during the COVID-19 pandemic period. This was seen both in the COVID-19 period group as a whole and in the hip fracture subgroup analysis. It was also noted that the length of stay was significantly shorter for the COVID-19 period group and the COVID-19 period hip fracture subgroup.

Mortality of patients with fragility fracture without COVID-19 was significantly higher among pandemic period admissions (14.7%) than in the pre-pandemic cohort (10.2%) after adjusting for age and sex (HR=1.86; 95% CI 1.41 to 2.45; p<0.001) (figures 1 and 2).

Table 3A shows variables that may have influenced outcomes. The cohort of COVID-19-positive patients was significantly older than the COVID-19-negative patients in the same time period. In relation to excess deaths in patients without COVID-19, table 3A also shows that the COVID-19-negative patients had a significantly shorter length of stay, lower frequency of operative treatment and less than expected ventilatory support than controls. There were no differences seen for sex or age between groups.

An analysis of patients with hip fracture revealed a mortality of 8.4% among 190 admissions in the pre-pandemic set, and of 15.48% among 168 pandemic admissions with no COVID-19 diagnosis, resulting in an

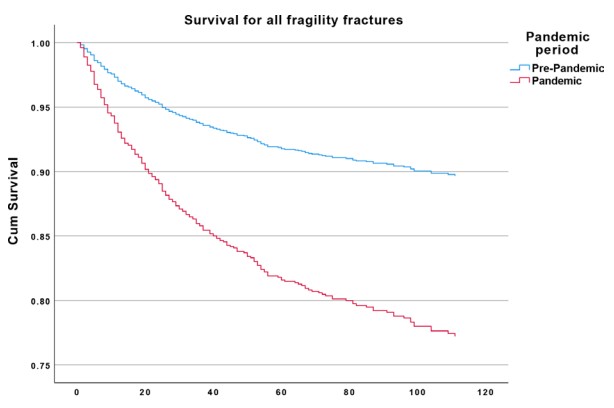

**Figure 1** Reduced survival in non-COVID-19 fragility fractures. Kaplan-Meier survival plot. Red: COVID-19 pandemic (COVID-19 negative) patients with fragility fracture. Blue: pre-COVID-19 pandemic patients with fragility fracture.

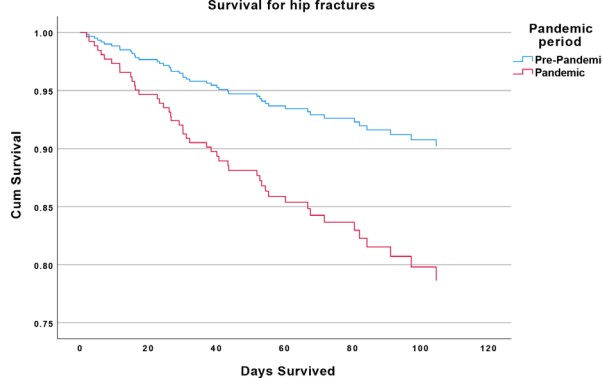

**Figure 2** Reduced survival in non-COVID-19 fragility hip fractures. Kaplan-Meier survival plot. Red: COVID-19 pandemic (COVID-19 negative) patients with fragility hip fracture. Blue: pre-COVID-19 pandemic patients with fragility hip fracture.

**Table 3** Complications, patient and treatment factors by group

| A | COVID-19 period group COVID-19 positive (total: n=104) | COVID-19 period group COVID-19 negative (total: n=728) | Control period group (total: n=1014) | P value |
|---|---|---|---|---|
| Age on admission, mean years (±SD); n | 81.0 (±11.9) | 73.9 (±20.7) | | 0.001* |
| | | 73.9 (±20.7) | 72.8 (±21.4) | 0.301* |
| Male sex, n (%) | 47/104 (45.2) | 292/728 (40.1) | | 0.338† |
| | | 292/728 (40.1%) | 385/1014 (38.0) | 0.370† |
| Length of stay, mean days (±SD); n | 14.8 (±10.6) | 8.0 (±8.3) | | <0.001‡ |
| | | 8.0 (±8.3) | 12.1 (±14.3) | <0.001‡ |
| Operative treatment, n (%) | 12/104 (11.5) | 58/728 (8.0) | | 0.255† |
| | | 58/728 (8.0) | 765/1014 (75.4) | <0.001† |
| Acute Respiratory Distress Syndrom (ARDS) or sepsis, n (%) | 5/104 (4.8) | 26/728 (3.6) | | 0.576† |
| | | 26/728 (3.6) | 26/1014 (2.6) | 0.254† |
| Ventilation post-procedure, n (%) | 6/104 (5.8) | 24/728 (3.3) | | 0.253† |
| | | 24/728 (3.3) | 54/1014 (5.3) | 0.046† |

| B | COVID-19 period hip fracture subgroup COVID-19 positive (total: n=12) | COVID-19 period hip fracture subgroup COVID-19 negative (total: n=168) | Control hip fracture subgroup (total: n=190) | P value |
|---|---|---|---|---|
| Age on admission, mean years (±SD); n | 82.5 (±6.3) | 83.1 (±9.0) | | 0.818* |
| | | 83.1 (±9.0) | 81.9 (±9.2) | 0.181† |
| Male sex, n (%) | 4/12 | 44/168 (26.1) | | 0.736† |
| | | 44/168 (26.1) | 51/190 (26.8) | 0.905† |
| Length of stay, mean days (±SD); n | 17.3 (±7.2) | 10.7 (±5.15) | | <0.001‡ |
| | | 10.7 (±5.15) | 15.3 (±7.4) | <0.001‡ |
| Hip fracture treatment | | | | |
| ► Arthroplasty | 8/12 | 68/165¶ (41.2%) | 81/190 (42.6%) | 0.227§ |
| ► Fixation | 4/12 | 90/165¶ (54.6%) | 103/190 (54.2%) | 0.355§ |
| ► Non-operative | 0/12 | 7/165¶ (4.2%) | 6/190 (3.2%) | 0.587§ |
| Pulmonary embolus or Deep Vein Thrombosis (DVT) | 0/12 | 0/168 | 2/190 (1.1%) | |
| Pneumonia | 3/12 | 14/168 (8.3%) | 13/190 (6.8%) | 0.081§ |
| Urinary Tract Infection (UTI) | 3/12 | 7/168 (4.2%) | | 0.021† |
| | | 7/168 (4.2%) | 10/190 (5.3%) | 0.804† |
| Cerbrovascular Accident (CVA) | 0/12 | 0/168 | 0/190 | |
| Myocardial Infarction (MI) or Acute Coronary Syndrom (ACS) | 0/12 | 2/168 (1.2%) | 0/190 | |
| Dislocation or failure of fixation | 0/12 | 0/168 | 1/190 (0.5%) | |
| Blood transfusion | 2/12 | 32/168 (19.1%) | 40/190 (21.1%) | 0.856§ |
| Renal failure | 2/12 | 18/168 (10.7%) | 22/190 (11.6%) | 0.813§ |
| Pressure ulcer | 0/12 | 1/168 (0.6%) | 1/190 (0.5%) | |
| *Clostridium difficile* | 0/12 | 0/168 | 0/190 | |
| Body Mass Index (BMI) | 24.7 (±3.5); 9 | 24.27 (±5.1); 150 | | 0.768* |
| | | 24.27 (±5.1); 150 | 23.2 (±4.4); 169 | 0.072† |

*Two-tailed distribution with homoscedastic variance Student's t-test.
†Fisher's exact test.
‡Kruskal-Wallis test.
§X$^2$ statistic calculated on a 2×3 contingency table.
¶Data were only available for 165 patients.
BMI, body mass index; MI, myocardial infarction; UTI, urinary tract infection.

HR=2.08 (95% CI 1.11 to 3.90; p=0.021). After further adjustment for clinical frailty scores, this became HR=2.15 (95% CI 1.15 to 4.04; p=0.0162).

## DISCUSSION

In this study, we report significantly higher mortality among patients with fragility fracture with no COVID-19-related diagnosis admitted during the COVID-19 pandemic period compared with patients admitted with the same ICD-10 codes to the same hospital in the same time period of the year 2019. We also report significantly shorter hospital length of stay during the months of the pandemic compared with the same period a year earlier. These results hold true after adjustment for potential confounders such as age and sex. A subanalysis on femoral neck fractures where frailty indices and other clinical assessments were readily available showed a similar pattern while there was no difference in frailty between the pandemic and pre-pandemic cohorts.

Limitations of this study include potential variables that have not been controlled that may have influenced the effects that have been seen. Knowledge of the patients with fragility fracture who contracted COVID-19 in the community, who subsequently died within the period of assessment, is not known and may account for a proportion of the excess mortality. In addition, it is known that during the first wave period of investigation in this study, the institutional compliance with the best practice tariff was reduced. There is the potential that the redeployment of staff during the COVID-19 first wave, particularly the orthogeriatric support and physiotherapists, for patients with fragility fracture, may have had a negative influence on outcomes. However, specific data about the effect of redeployment as a variable are not available for this study.

A recent report from the Office for National Statistics found that England had the highest overall relative excess mortality out of all the European countries compared during the first 6 months of 2020.[12] While none of the four UK nations had a peak mortality level as high as Spain or the worst-hit local areas of Spain and Italy, excess mortality was geographically widespread throughout the UK during the pandemic, whereas it was more geographically localised in most countries of Western Europe. Our data add to this picture by showing that excess mortality involves patients without COVID-19 and areas of England with rates of infection below the national average. Achievement of best practice hip fracture tariff[13] at our institution during the pandemic fell from an average of 42% (March–June 2019) to 38% (March–June 2020) with a particular failure to achieve the delirium assessment, which fell from an average of 96% to 62% for the respective periods. In addition, there was a significant decrease in the number of patients with hip fracture who were not delirious after surgery, from 81% to 63%, respectively. This indicates breakdown of standard care pathways due to extrinsic pressures and provides a likely explanation for the excess mortality. As there are no agreed pathways or metrics for other fractures, it is impossible to prove, although probable that the same factors apply.

Hip fractures occurred at the same frequency during the COVID-19 period and the national lockdown, as before it, with no differences identified for age, sex, BMI, AMTS, mobility, NHFS, anaemia, fracture type or frailty between groups. Similar incidence pre-COVID-19 and post-COVID-19 with similar demographics allows a valid comparison to examine excess deaths.

Analysis of the COVID-19-negative subset of those presenting during the COVID-19 period compared with the control period does identify associations that may be relevant. While there were no differences identified between these groups for age, the development of Acute Respiratory Distress Syndrome (ARDS) or sepsis of the requirement for ventilatory support post-procedure, significant difference was noted in the COVID-19 period group (COVID-19-negative subset) for a shorter length of stay and reduced number of procedures. Both differences suggest that the standard care pathways were altered for fragility fractures, which is further supported by the observed fall in Best Practice Tarrif (BPT) achievement. A reduced length of stay is often considered a positive outcome. However, if the threshold for a safe discharge is altered, for instance in the face of a great need for more inpatient beds to manage a pandemic, there may be unintended consequences affecting mortality. A significant proportion of patients with fragility fractures require input from health and social care in the form of rehab services and residential homes. During the pandemic period, it was unknown regarding the virulence of spread of COVID-19 transmission within care homes and limiting testing in this group of people was available; thus, this may have also influenced our mortality findings.

For those presenting with hip fractures during the COVID-19 period, who did not test positive for COVID-19, that no differences were seen for age, sex, BMI, treatment type (including non-operative treatment) and complications including: venous thromboembolism, urinary tract infection, stroke, myocardial infarction, failure of fixation or dislocation, transfusion, acute renal failure, pulmonary embolus or *Clostridium difficile* when compared with the control group suggests that understanding the reasons for the excess mortality is complex. Despite this, reduction in length of stay seems to be a significant factor in the hip fracture subgroup analysis, where changes to operative treatment frequency (as for the fragility fracture group as a whole) are not. It has been demonstrated that a COVID-19 infection independently causes an increased mortality compared with

those presenting with hip fractures who do not contract COVID-19.[14–16]

During the COVID-19 period, only 12 patients tested positive for COVID-19. When compared with those presenting during the same time period, they were older and had a longer length of stay. Age, as a risk factor for developing COVID-19, is well established.[17]

During the COVID-19 pandemic period, services were subjected to changes in both personnel and patient pathways. Although one may assume that there would have been a reduction in staff due to an increase in sickness and testing positive for COVID-19, the measures put into place did not impact the staffing levels and actually improved the staffing levels. Although at the expense of the following, trainee doctors did not rotate to other specialties or units to maintain continuity of care, clinicians out of programme for research/experience and research nurses were brought back into clinical practice, doctors were placed onto megarotas and staff normally working in the elective setting were moved to the acute setting to improve patient flow. The change in the established evidence-based guidelines for patient care pathways is altered to help improve patient flow within the hospital, facilitate discharge and improve capacity for the at time unknown effect of the pandemic. These were approved both regionally and nationally by subspecialty working groups.[18]

These data suggest that excess deaths are likely due to changing established, evidence-based care pathways, particularly regarding thresholds for safe discharge and length of stay. Excess mortality seen in this vulnerable population with fragility fracture is likely to be minimised by maintaining standards of care during further waves of the COVID-19 or future pandemics.

**Author affiliations**
[1] Academic Orthopaedics, Trauma and Sports Medicine, University of Nottingham School of Medicine, Nottingham, UK
[2] Queens Medical Centre, Nottingham University Hospitals NHS Trust, Nottingham, UK
[3] Queen Elizabeth Hospital King's Lynn NHS Foundation Trust, King's Lynn, UK
[4] School of Medicine, University of Nottingham, Nottingham, UK
[5] NIHR Nottingham Biomedical Research Centre, Nottingham, UK

**Contributors** AI—methodology, data analysis and interpretation, original draft preparation, review, editing and revision of manuscript. AN—methodology, data analysis and interpretation, original draft preparation review and editing. LO—hospital coding, database manager, data extraction and acquisition and resources. JN—database manager, data extraction and acquisition, resources and review of manuscript. AV—conceptualisation of study, methodology, securing funding, interpretation of data, review of manuscript, approval of the version to be published and supervision of the study. BJO—conceptualisation of the study, methodology, interpretation of data, review of manuscript, approval of the version to be published and supervision of the study, guarantor.

**Funding** The research leading to these results received funding from UKRI/MRC Rapid Response COVID trial (grant number: MR/V027883/1).

**Competing interests** None declared.

**Patient and public involvement** Patients and/or the public were not involved in the design, or conduct, or reporting, or dissemination plans of this research.

**Patient consent for publication** Not required.

**Ethics approval** The study was registered and approved with our local hospital audit department (no. 20-333-C). This study did not require ethical approval, as we used anonymised data that were collected as part of routine care for clinical audit.

**Provenance and peer review** Not commissioned; externally peer reviewed.

**Data availability statement** Data are available upon reasonable request. The datasets generated during and/or analysed during the current study are not publicly available due to them containing patient-identifiable data within the NHS. But anonymised data are available from the corresponding author on reasonable request.

**ORCID iDs**
Adeel Ikram http://orcid.org/0000-0002-7520-6949
Alan Norrish http://orcid.org/0000-0003-3735-1042
Jessica Nightingale http://orcid.org/0000-0001-8359-5705
Ana Valdes http://orcid.org/0000-0003-1141-4471
Benjamin J Ollivere http://orcid.org/0000-0002-1410-1756

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
