## [Reviewer comments · BMJ Open]

ARTICLE DETAILS

TITLE (PROVISIONAL)	Has a change in established care pathways during the 1st wave of the Covid-19 pandemic led to an excess death rate in the fragility fracture population? A longitudinal cohort study of 1846 patients.
AUTHORS	Ikram, Adeel; Norrish, Alan; Ollivere, Luke; Nightingale, Jessica; Valdes, A; Ollivere, Benjamin

VERSION 1 – REVIEW

REVIEWER	Tóth, Mathias Darent Valley Hospital, Ageing & Health
REVIEW RETURNED	28-Nov-2021

GENERAL COMMENTS	Page 7 from line 8 onwards: the pandemic was declared in March 2020, not 2019, and the chosen study period was presumably chosen from 1st March 2020. Covid testing was notoriously unreliable during the first peak of the pandemic, and it has been estimated that at least 30% of tests were false negative (BMJ 2021;372:n287) We therefore moved away from relying on PCR testing and started making a 'clinical diagnosis'. It has not been made clear at what stage of the admission the Covid negative status was confirmed or how often this had been repeated during the hospital stay. Had study subjects been tested for their Covid status post discharge?
---

REVIEWER	Ong, Terence University of Malaya
REVIEW RETURNED	30-Nov-2021

GENERAL COMMENTS	It is a sobering read and is a further reminder (not that we need it) of how Covid19 has affected patients and the healthcare system that looks after them. Some clarification required from me - The choice of ICD-10 codes. Why only include S72 (femur fracture) and not the other codes pertaining to fractures affecting, eg pelvis, upper limbs, lower limbs?- 80% of patients were non-hip fractures. I wonder if a break down of these fractures are readily available? Different fracture carry with it different prognosis and there have been some centres that have reported different fractures presenting to clinical attention during Covid19 compared to non-Covid19 times.- Would the authors be able to share how their service was impacted during this time? Was there an increase in patient:nurse ratio? Medical staffing numbers? Lower WTE AHP input?- Covid19 testing strategy during this time. Were all admission routinely screened for Covid or was testing targeted at those symptomatic? Was there re-testing during patients' inpatient stay? In
--

	the early days of the pandemic, there was a huge reliance on respiratory symptoms to diagnose Covid19. Hence, I wonder if there could have been undiagnosed Covid19 that drove the excess mortality reported here.  - The Clinical Frailty Scale is an ordinal scale. I would suggest presenting and comparing it as such in the different analyses conducted, or into categories of not, mild, moderate or severe frailty if the numbers allow. - Outcomes were adjusted according to age and gender. If differences exist for types of fracture and frailty, it would be interesting to see if outcomes were any different. - Analysis for difference in length of stay used Student's T-test. The T-test assumes normal distribution of the continuous data. Length of stay tends to be (positively) skewed. - Did time to hip fracture operation change during Covid19? Is there available data on time from injury to presentation and if this was longer during the pandemic as pre-hospital care could have influenced the outcomes reported here? - Table 1a. Types of hip fracture and % not aligned in the table. - The number of Covid19 hip fractures in Table 2 is different from Table 1b - Line 55 page 7. Subanalysis was not just neck of femurs
--	---

REVIEWER	Raheman, Firas Lister Hospital
REVIEW RETURNED	03-Feb-2022

GENERAL COMMENTS	Interesting and Highly Relevant topic. This paper highlights the indirect impact of the COVID-19 pandemic on the hip fracture population. Manuscript of very high quality, with valid and statistically proven conclusions. Correct the study design and methodology, as well as the statistical analysis. The same valutations can be done for images, tables and any supplementary material. Accept for publication
--

VERSION 1 – AUTHOR RESPONSE

Reviewer 1 Comments	Thank you for your comments	
1. Page 7 from line 8 onwards: the pandemic was declared in March 2020, not 2019, and the chosen study period was presumably chosen from 1st March 2020.	We ask the reviewer to note the following in the text “Two time periods were selected to define groups: (i) the study group were admitted during the 1st wave of the UK COVID-19 pandemic, between 1 March 2020 to 1 June 2020, and discharged by 30 June 2020 (ii) the control group were admitted between 1 March 2019 to 1 June 2019, and discharged by 30 June 2019.” We believe you may have misunderstood that the control group was recruited from 1st March 2019 to 1st June 2019. We hope this satisfies this comment.	Materials and Methods

1. Covid testing was notoriously unreliable during the first peak of the pandemic, and it has been estimated that at least 30% of tests were false negative (BMJ 2021;372:n287) We therefore moved away from relying on PCR testing and started making a 'clinical diagnosis'. It has not been made clear at what stage of the admission the Covid negative status was confirmed or how often this had been repeated during the hospital stay. Had study subjects been tested for their Covid status post discharge?	All patients in the control group would not have had covid testing since this was in the pre pandemic period, we therefore assume that all patients in this cohort are Covid-19 negative. For the patients in the study group (pandemic period) all patients received a covid test on admission please note in the material and methods section “COVID status (as identified in the medical records as “Coronavirus SARS-COV-2” positive patients or with a positive reverse transcriptase polymerase chain reaction; rPCR)” Covid testing was repeated every 3 days as according to guidelines. Post discharge covid testing was not performed as these patients either ended up home or institutionalised. Pre-discharge covid testing was performed as part of routine inpatient testing to identify potential carriers to permit the required isolation period before returning to potential household or care home contacts. We have therefore included the following as an additional statement in the material and methods section. “Covid testing was performed on admission for all patients and repeated every 3 days during inpatient stay, this was to facilitate and maintain safe distancing and isolation for covid contacts. Post discharge testing was not performed as these patients were either discharged home or to an institution. Hip fracture patients are not routinely followed up and those patients with other fragility fractures who are, were advised to not attend follow up if they developed symptoms or had a positive test.” We hope this satisfies this comment	Materials and Methods
Reviewer 2 Comments	Thank you for your comments	
1. It is a sobering read and is a further reminder (not that we need it) of how Covid19 has affected patients and the healthcare system that looks after them. Some clarification required from me - The choice of ICD-10 codes. Why only include S72 (femur fracture) and not the other codes pertaining to fractures affecting, eg pelvis, upper limbs, lower limbs?	Our choice of ICD-10 codes was used to identify those patients specifically with fragility fractures i.e., fall from standing height. An inclusion of all fractures would include patients who sustained high energy trauma, paediatric patients, patients with different mechanism of injury and these injuries can be managed differently in different units. This would increase the heterogeneity of the results. By using the ICD-10 codes we have, we are able to identify those patients who are elderly with significant medical co-morbidities the patient population in whom Covid-19 affected severely and in whom evidenced based hospital pathways have been formulated to try and improve clinical outcomes in an already at-risk population. Being specific with S72 femur	Materials and methods

	fracture, we would be able to provide a more precise recommendation following our analysis of the results with regards to changing of patient care pathways. We have therefore elected not to change the ICD-10 codes as we feel these codes capture this group of patients well and still provides a useful message to disseminate to the wider audience of BMJ Open readers. We hope this satisfies this comment.	
1. 80% of patients were non-hip fractures. I wonder if a breakdown of these fractures are readily available? Different fracture carry with it different prognosis and there have been some centres that have reported different fractures presenting to clinical attention during Covid19 compared to non-Covid19 times.	We appreciate that there are differences in outcomes depending on injury pattern and location. Unfortunately, we do not have the data to go into each individual injury. Many injuries although roughly managed similarly across the country are not subject to strict national guidelines and therefore would be difficult to compare amongst regions and countries. The hip fracture population are managed ubiquitously as the standards are set out by the National Institute of Clinical Excellence (NICE). We therefore provided this group as a subgroup analysis. This group of patients has a significant morbidity and mortality even prior to the effect of the pandemic on care pathways and any change in established care pathway we would presume would affect this group more sensitively. We have added the following to the materials and methods section to highlight the above explanation. “These ICD-10 codes were selected to capture patients who were likely presenting with fragility fractures, and were identified from data entered for the national Hospital Episode Statistics submission. Patients who present with fragility fractures have significant medical comorbidities, are elderly and frail. This group of patients are the ones at greatest risk of both Covid-19 and any alterations to established evidenced-based care pathways.” We hope this satisfies this comment.	
1. Would the authors be able to share how their service was impacted during this time? Was there an increase in patient: nurse ratio? Medical staffing numbers? Lower WTE AHP input? - Covid19 testing strategy during this time. Were all admission routinely screened for Covid or	Our service was subject to changes in both the personnel and management of patient pathways. During the pandemic there was an increase in staff members off sick or diagnosed with Covid-19, however to maintain ratios a number of measures were introduced which included the following. In some instances, there was an increase in the number of staff members present. 1. Trainee doctors did not rotate to their next placement but stayed	

was testing targeted at those symptomatic? Was there re-testing during patients' inpatient stay? In the early days of the pandemic, there was a huge reliance on respiratory symptoms to diagnose Covid19. Hence, I wonder if there could have been undiagnosed Covid19 that drove the excess mortality reported here.	on with their current role to provide continuity of care  2. Doctors out of training for experience, & research were brought back to clinical practice to aid with the staffing levels 3. Doctors were placed onto mega rotas, seconded to acute specialties and intensive care where appropriate and clinical need arose. 4. Clinical research nurses were brought into clinical practice to help address staffing issues. 5. Staff normally working in the elective setting were seconded to the acute setting to also help address staffing issues and improve patient flow. We have therefore added the following into the discussion “During the Covid-19 pandemic period services were subjected to changes in both personnel and patient pathways. Although one may assume that there would have been a reduction in staff due to an increase in sickness and testing positive for Covid-19, the measures put into place did not impact the staffing levels and actually improved the staffing levels. Albeit at the expense of the following; trainee doctors did not rotate to other specialties or units to maintain continuity of care, clinicians out of programme for research / experience and research nurses were brought back into clinical practice, doctors were placed onto megarotas and staff normally working in the elective setting were moved to the acute setting to improve patient flow. The change in the established evidenced based guidelines for patient care pathways were altered to help improve patient flow within the hospital, facilitate discharge and improve capacity for the at time unknown effect of the pandemic. These were approved both regionally and nationally by subspecialty working groups.” Covid testing was performed on admission and then every three days while an inpatient. “Covid testing was performed on admission for all patients and repeated every 3 days during inpatient stay, this was to facilitate maintain safe distancing and isolation for covid contacts. Post discharge testing was not performed as these patients were either discharged home or to an institution. Hip fracture patients are not routinely followed up and those patients with other fragility fractures who are were advised to not attend follow up if they developed	Discussion Materials and methods
---	---	--

	symptoms or had a positive test.” We hope the above satisfies these comments.	
1. The Clinical Frailty Scale is an ordinal scale. I would suggest presenting and comparing it as such in the different analyses conducted, or into categories of not, mild, moderate or severe frailty if the numbers allow.	We acknowledge that the Clinical Frailty Scale (CFS) score is an ordinal scale. In table 1 and 2 we have included the CFS to see if there is a difference in frailty between the control and study groups which may have explained the difference in outcomes. Our results demonstrate there is no difference in mean frailty. Had there been, we would have recorded this as an ordinal scale to show where the difference occurred. We have therefore elected not to change this in our manuscript, however if the editor wishes we could amend this if it were deemed imperative to do so. We hope the above satisfies this comment.	
1. Outcomes were adjusted according to age and gender. If differences exist for types of fracture and frailty, it would be interesting to see if outcomes were any different.	We acknowledge this comment, unfortunately to capture each fracture subtype within the remit of fragility fractures would have been a too exhaustive task. We hope the above satisfies this comment.	
1. Analysis for difference in length of stay used Student's T-test. The T-test assumes normal distribution of the continuous data. Length of stay tends to be (positively) skewed.	We acknowledge that length of stay tends to be positively skewed, and the Students T-test assumes normal distribution. Therefore, we have used the Kruskal-Wallis test (non-parametric test) and re run the analysis. We have made the amendments to table 2 and 3. We hope the above satisfies this comment.	Table 2 and table 3
1. Did time to hip fracture operation change during Covid19? Is there available data on time from injury to presentation and if this was longer during the pandemic as pre-hospital care could have influenced the outcomes reported here?	Hip fractures were still managed according to best practice tariff and NICE guidelines, unfortunately data on time of injury to presentation was not collected and therefore we are unable to add this into the analysis for this study. We hope the above satisfies this comment.	
1. Table 1a. Types of hip fracture and % not aligned in the table.	We have formatted the table to ensure correct alignment We hope this satisfies this comment	Results table 1
1. The number of Covid19	We have corrected the number in table 2	Results

hip fractures in Table 2 is different from Table 1b	We hope this satisfies this comment	table 2
1. Line 55 page 7. Subanalysis was not just neck of femurs	We have corrected this to "An analysis of hip fracture patients" We hope this satisfies this comment	Results section
Reviewer 3 Comments		
Interesting and Highly Relevant topic. This paper highlights the indirect impact of the COVID-19 pandemic on the hip fracture population. Manuscript of very high quality, with valid and statistically proven conclusions. Correct the study design and methodology, as well as the statistical analysis. The same valutations can be done for images, tables and any supplementary material. Accept for publication	Thank you for your comments